# Impact of Diabetes on Cardiac Function in Patients with High Blood Pressure

**DOI:** 10.3390/ijerph18126553

**Published:** 2021-06-18

**Authors:** Nabila Soufi Taleb Bendiab, Souhila Ouabdesselam, Latefa Henaoui, Marilucy Lopez-Sublet, Jean-Jacques Monsuez, Salim Benkhedda

**Affiliations:** 1Department of Cardiology, Faculty of Medicine Aboubekr Belkaid, University Hospital Tlemcen, Tlemcen 13000, Algeria; sarra13dz@yahoo.fr; 2Department of Cardiology, Mustapha University Hospital Center Algiers, Algiers 16000, Algeria; souhila85@hotmail.com (S.O.); sbenkhedda@gmail.com (S.B.); 3Cardiology Oncology Research Collaborative Group (CORCG), Faculty of Medicine BENYOUCEF BENKHEDDA University, Algiers 16000, Algeria; 4Department of Epidemiology, Faculty of Medicine Aboubekr Belkaid, University Hospital Tlemcen, Tlemcen 13000, Algeria; henaouilatifa@yahoo.fr; 5APHP Hôpital R Muret, Hôpitaux Universitaires de Paris Seine Saint Denis, 93270 Sevran, France; marilucy.lopez-sublet@aphp.fr; 6Centre d’HTA, Hôpital Avicenne,93000 Bobigny, France

**Keywords:** diabetes, hypertension, left ventricular function, global longitudinal strain

## Abstract

*Background*: Although the combination of high blood pressure (HBP) and type 2 diabetes (T2DM) increases the risk of left ventricular (LV) dysfunction, the impact of T2DM on LV geometry and subclinical dysfunction in hypertensive patients and normal ejection fraction (EF) has been infrequently evaluated. *Methods*: Hypertensive patients with or without T2DM underwent cardiac echocardiography coupled with LV global longitudinal strain (GLS) assessment. *Results*: Among 200 patients with HBP (mean age 61.7 ± 9.7 years) and EF > 55%, 93 had associated T2DM. Patients with T2DM had a higher body mass index (29.9 ± 5.1 kg/m^2^ vs. 29.3 ± 4.7 kg/m^2^, *p* = 0.025), higher BP levels (158 ± 23/95 ± 13 vs. 142 ± 33/87 ± 12 mmHg, *p* = 0.003), a higher LV mass index (115.8 ± 32.4 vs. 112.0 ± 24.7 g/m^2^, *p* = 0.004), and higher relative wall thickness (0.51 ± 0.16 vs. 0.46 ± 0.12, *p* = 0.0001). They had more frequently concentric remodeling (20.4% vs. 16.8%, *p* < 0.001), concentric hypertrophy (53.7% vs. 48.6%, *p* < 0.001), elevated filling pressures (25.8 vs. 12.1%, *p* = 0.0001), indexed left atrial volumes greater than 28 mL/m^2^ (17.2 vs. 11.2%, *p* = 0.001), and a reduced GLS less than −18% (74.2 vs. 47.7%, *p* < 0.0001). After adjustment for BP and BMI, T2DM remains an independent determinant factor for GLS decline (OR = 2.26, 95% CI 1.11–4.61, *p* = 0.023). *Conclusions:* Left ventricular geometry and subclinical LV function as assessed with GLS are more impaired in hypertensive patients with than without T2DM. Preventive approaches to control BMI and risk of T2DM in hypertensive patients should be emphasized.

## 1. Introduction

The colluding effects of the two most common noncommunicable diseases, hypertension and type 2 diabetes mellitus (T2DM), result in a severalfold elevation in risk for cardiovascular morbidity and mortality [1]. Hypertension is associated with left ventricular (LV) remodeling [2]. Pressure overload results in increased wall thickness and LV mass, and, in turn, in diastolic dysfunction resulting from collagen deposition [1,2]. Although effective control of HBP allows most patients to remain asymptomatic for many years, LV hypertrophy, diastolic dysfunction, and heart failure may develop overtime, suggesting that early recognition of subclinical cardiac damage may be useful [2,3].

The prevalence of hypertension is significantly higher in diabetic patients, as high as 50–70% in some studies, and, reciprocally, T2DM is commonly associated in hypertensive patients [4]. In addition, diabetes also impairs LV structure and function, leading to diastolic dysfunction and decreased LV function [5].

The association of HBP and T2DM further increases the risk of progressive LV dysfunction and heart failure [6]. Although previous studies have observed more severe subclinical systolic dysfunction in patients with coexisting T2DM and hypertension than in patients with T2DM only [7,8], none have evaluated the impact of T2DM on LV geometry and LV subclinical dysfunction in patients with hypertension.

Although two-dimensional echocardiography (2-DE) may track changes in LV diastolic and systolic function, assessment of LV global longitudinal strain (GLS) by speckle-tracking provides recognition of LV impairment and its associated factors at an earlier stage [9].

This study investigates the impact on subclinical cardiac function of concomitant diabetes in a cohort of patients with HBP.

## 2. Methods

### 2.1. Patients

This cross-sectional study was conducted in 200 hypertensive patients aged ≥30 years and followed up in an outpatient clinic between January 2018 and January 2019. There were 93 patients with and 107 without concomitant T2DM. All had normal LV ejection fraction (EF) defined as EF > 55%. Hypertension was defined by a systolic blood pressure (BP) ≥ 140 mmHg, a diastolic BP ≥ 90 mmHg, or both, or current treatment for HBP. Patients with secondary hypertension, concomitant cardiomyopathy, heart failure, moderate to severe valve heart disease, coronary artery disease, atrial fibrillation, or paced cardiac rhythm were not included.

The study was approved by the committee for ethics of the Mustapha hospital. All patients gave formal informed consent.

### 2.2. Clinical Assessment

Patients’ past medical history, including cardiovascular risk factors, duration, treatment and control of HBP, as well as baseline clinical characteristics, including age, gender, weight, height, and body mass index (BMI) were recorded. HBP was classified using the European Society of Hypertension and the European Society of Cardiology guidelines [10,11]. Brachial BP was measured using a sphygmomanometer cuff after a 5 min rest period in a seated position before echocardiographic examination. The mean of two consecutive readings was calculated and retained for analysis.

Blood sampling included glucose and creatinine level determinations. Creatinine clearance was calculated using the MDRD formula. Lipid profile determination included total cholesterol, LDL-cholesterol, HDL-cholesterol, and triglyceride levels.

LVH was defined using the Sokolow-Lyon index obtained from a resting 12-lead electrocardiogram examination.

### 2.3. Echocardiographic Assessment

Ultrasound examination included 2DE imaging, M-mode evaluation, a Doppler-coupled analysis using pulsed, continuous, and color methods, and a 2-dimensional strain measurement (2D strain) using a harmonic 4.0 MHz variable-frequency phased-array transducer VIVID S6 ultrasound system (General Electric Healthcare, Milwaukee, WI, USA). The same experienced physician performed all examinations with the patient in the left lateral position along the parasternal long-axis view, short axis, and apical 2, 3, and 4 cavities. Records connected to an ECG were obtained from 3 consecutive cardiac cycles at each plane at end-expiration breath, as recommended by the American Society of Echocardiography [12].

#### 2.3.1. Cardiac Geometry

M-mode derived measurements of end-diastolic LV diameter (Dd), end-systolic LV diameter (Ds), end-diastolic thickness of the ventricular septum (VST), and end-diastolic thickness of the LV posterior wall (PWT) allowed calculation of LV mass and relative LV wall thickness (RWT) as follows:-LV mass (g) = 0.8 × 1.04 × [(Dd + PWT + VST)^3^ − Dd^3^] + 0.6-RWT = 2 × PWT/Dd

LV mass was indexed to body surface area (LVMI). Left ventricular hypertrophy was defined by an LVMI > 115 g/m^2^ in men and >95 g/m^2^ in women, except for obese patients in whom LVH was defined as >49g/m^2.7^ in men and >45g/m^2.7^ in women [8,9,10,11].

According to the ASE guidelines, cardiac geometry of patients was classified using RWT and LVMI as follows: (1) normal geometry (normal RWT and LVMI); (2) concentric remodeling (normal LVMI, increased RWT); (3) concentric hypertrophy (RWT and LVMI both increased); or (4) eccentric hypertrophy (normal RWT, increased LVMI) [7]. Left atrial (LA) end-systolic volume was also measured.

#### 2.3.2. Cardiac Function

The Simpson’s 2-chambers and 4-chambers biplane method was applied to measure LV ejection fraction (EF). Transmitral flow velocities measured by pulsed Doppler were used to assess LV diastolic function, including peak early diastolic velocity (E), deceleration time from the peak of the early diastolic wave to baseline, peak atrial systolic velocity (A), and the E/A ratio. Velocities of mitral annular motion were measured at the LV septal and lateral annulus by pulsed tissue Doppler assessment, allowing calculation of peak early diastolic motion velocity (E’), mean E’ (E’septal + E’lateral/2), and the ratio E/E’.

The maximum velocity of the tricuspid regurgitation jet allowed calculation of pulmonary artery systolic pressure.

### 2.4. Two-Dimensional Strain Imaging

Using the speckle tracking technique, velocities, strain curves, and systolic longitudinal strain were measured from apical long-axis views at end-expiration breath of 3 consecutive cardiac cycles, allowing determination of the global longitudinal peak systolic strain (GLS) to assess LV ventricular function [9].

## 3. Statistical Analysis

Comparisons of continuous variables expressed as mean values ± standard deviation (SD) were performed using ANOVA. Comparisons of categorical variables were performed using the chi-square test or the Fisher exact test when appropriate. Data analysis was performed using SSPS software (Graduate Pack for Windows, version 20) (IBM, Armonk, NY, USA). A *p*-value < 0.05 was considered significant.

## 4. Results

### 4.1. Baseline Characteristics

Mean age of the 200 hypertensive patients (39% men, 61% women) included was 61.7 ± 9.7 years. In most (71%), HBP lasted for at least 5 years. All patients were of the same Northern African ethnicity, born and living in Algeria. BMI was 29.9 ± 5.1 kg/m^2^ in diabetics versus 29.3 ± 4.7 kg/m^2^ in patients without T2DM (*p* = 0.025). Systolic blood pressure was slightly higher in diabetics versus nondiabetic patients. Baseline clinical characteristics of the study population are listed in Table 1.

### 4.2. Cardiac Geometry and Function

Echocardiographic findings of the patients are listed in Table 2. LVH was observed in 65% of patients with and 61% of those without T2DM, with a mean LVMI of 115.8 ± 32.4 versus 112.0 ± 24.7 g/m^2^, respectively (*p* = 0.004). Concentric remodeling was found in 20.4% versus 16.8% of patients with and without T2DM, and concentric hypertrophy in 53.7% versus 48.6%, respectively (*p* < 0.001 for both).

RWT was >0.42 in 144 patients (72%). Mean RWT was higher in patients with than without T2DM (0.51 ± 0.16 vs. 0.46 ± 0.12, *p* = 0.0001).

An impaired diastolic function was seen in 178 (89%) patients. Elevated filling pressures were observed in 24 patients with and 13 patients without T2DM (25.8 vs. 12.1%, *p* = 0.0001).

Indexed left atrial volumes *>* 28 mL/m^2^ were observed in 16 patients with and 12 patients without T2DM (17.2 vs. 11.2%, *p* = 0.001).

### 4.3. Global Myocardial Strain

Mean GLS values obtained from apical 3- and 4-chamber views were lower in hypertensive patients with than in those without T2DM (Table 3). A reduced GLS less than −18% was observed in 69 patients with and 51 patients without T2DM (74.2 vs. 47.7%, *p* < 0.0001).

### 4.4. Multivariate Analysis

Although hypertensive patients with T2DM have higher blood pressure levels and higher BMI than those without T2DM, after adjustment for blood pressure and BMI using ANOVA, T2DM remains an independent determinant factor for GLS decline (Table 3 and Table 4). Independent factors of lower GLS risk resulting from logistic regression in hypertensive patients are listed in Table 5. They include non-control of blood pressure, BMI, diabetes, and RWT.

### 4.5. Impact of HBP Treatment

Among the four classes of hypertensive drugs administered to patients (angiotensin-converting enzyme inhibitors, diuretics, beta-blocking agents, calcium channel inhibitors), the angiotensin-converting enzyme inhibitors provided a protective effect in patients with HBP and T2DM (Table 6).

## 5. Discussion

A decreased GLS has been reported previously in 15% to 42% of patients with HBP, depending on both severity and control of HBP [13,14,15,16]. In addition, GLS is more reduced in patients with long-lasting HBP, overweight, metabolic changes, and diabetes [16]. On the other hand, early impairment of GLS is observed in patients with T2DM [5,17], which is worsened by coexistent hypertension [18,19,20].

The present study shows two main results: despite a conserved LV ejection fraction, (1) LV geometry is more impaired in hypertensive patients with than in those without T2DM. The former are more prone to concentric remodeling and concentric hypertrophy than the latter; (2) coexisting T2DM further deteriorates LV subclinical function as assessed with GLS in patients with HBP.

These findings are consistent with those reported by Li et al., using cardiac magnetic resonance to assess GLS in hypertensive patients with or without T2DM [21]. Both indicate the deleterious effect of T2DM on myocardial function in patients with hypertension. Such a subclinical impairment should be detected as early as possible to optimize treatment and blood pressure control in these high-risk patients.

The mechanisms by which T2DM further impairs subclinical LV function in hypertensive patients remain unclear. Factors associated with impaired geometry and decreased GLS in patients with hypertension include duration of hypertension, uncontrolled hypertension, LV hypertrophy, overweight, and related metabolic changes [16]. More specific effects of diabetes have also been reported to contribute to LV dysfunction. Direct effects of insulin resistance/hyperglycemia on the myocardium may contribute to a condition generally referred to as diabetic cardiomyopathy, including mitochondrial dysfunction, endoplasmic reticulum stress, oxidative stress, production of advanced glycosylation end products, impaired calcium homeostasis, renin–angiotensin–aldosterone system activation, and microvascular dysfunction [5]. A recent MRI study showed a significant correlation between subclinical LV dysfunction and impaired myocardial perfusion in hypertensive patients with diabetes. Abnormalities of the microvascular perfusion may compromise nutrients and oxygen delivery, energy production, and, in turn, myocardial contractility [21].

### 5.1. Preventive Implications

Since simultaneous occurrence of T2DM further impacts myocardial function of hypertensive patients, overweight and subsequent metabolic changes should be avoided among them. Optimal control of glycemia and control of cardiovascular risk factors, especially HBP, are associated with a decrease in all-cause mortality and cardiovascular mortality [22]. Indeed, weight loss has been reported to improve cardiac structure and function of patients with overweight [23]. Accordingly, effective and optimal treatment of HBP should be obtained in very high risk hypertensive patients such as those with T2DM [24]. In addition, antihypertensive drugs such as angiotensin-converting enzyme inhibitors, as observed in the present study, may have direct protective effects, and their use may contribute to reduced vascular complications in patients with HBP and concomitant T2DM [25].

### 5.2. Limitations of the Study

A first limitation pertains to the evaluation of subclinical LV function using GLS only. This was done according to the fact that patients included were free of any cardiac disease other than HBP, including coronary artery disease and regional wall motion abnormality. A second limitation results from the fact that wall stress impacts GLS. However, in the multivariable analysis of our study, T2DM and RWT remained independent risk factors for GLS decline.

## 6. Conclusions

Left ventricular geometry and subclinical LV function measured with GLS are more impaired in hypertensive patients with than in those without T2DM. Preventive approaches to control BMI and risk of T2DM in hypertensive patients should be emphasized.

## Figures and Tables

**Table 1 ijerph-18-06553-t001:** General characteristics of the study population.

	Hypertension and T2DM, *n* = 93	Hypertension without T2DM, *n* = 107	*p*-Value
Age, years	61 ± 9.7	62 ± 6.4	0.64
M/F	0.57	0.87	0.88
Body mass index, kg/m^2^	29.9 ± 5.1	29.3 ± 4.7	0.025
Systolic BP, mmHg	158 ± 23	142 ±33	0.002
Diastolic BP, mmHg	95 ± 13	87 ± 12	0.003
Chronic kidney disease, *n*	13 (13.9%)	9 (8.4%)	0.003
Dyslipidemia, *n*	27 (29%)	28 (26.1%)	0.46

BP: blood pressure; M/F: men/women; T2DM: type 2 diabetes.

**Table 2 ijerph-18-06553-t002:** Echocardiographic characteristics of the study population.

	Hypertension with T2DM, *n* = 93	Hypertension without TDM, *n* = 107	*p*-Value
LA, cm	39.4 ± 5.6	38.6 ± 5.8	0.25
LA volume index mL/m^2^	17.4 ± 7.4	6.5 ± 7.0	0.004
Relative wall thickness	0.51 ± 0.16	0.46 ± 0.12	0.0001
LV mass index, g/m^2^	115.8 ± 32.4	112.0 ± 24.7	0.007
LV ejection fraction, %	61.4 ± 6.0	61.6 ± 6.3	0.63
Midwall fractional shortening, %	35.1 ± 7.1	35.0 ± 6.6	0.45
Normal LV geometry, *n*	9 (9.7%)	25 (23.3%)	0.0003
Concentric remodeling, *n*	19 (20.4%)	18 (16.9%)	<0.0001
Concentric hypertrophy, *n*	50 (53.7%)	52 (48.6)	<0.0001
Eccentric hypertrophy, *n*	15 (16.1%)	12 (11.2%)	0.002
E/A ratio	0.8 ± 0.5	0.7 ± 0.3	0.25
E/E’ ratio	6.8 ± 2.7	6.4 ± 2.4	<0.0001
GLS, %	−16.4 ± 3.0	−17.8 ± 3.2	<0.0001

E/A: early to late mitral filling velocity ratio; E/E’: average mitral-to-peal early diastolic annular ratio; GLS: global longitudinal strain; LA: left atrial; LV: left ventricular; T2DM: type 2 diabetes.

**Table 3 ijerph-18-06553-t003:** Two-way analysis of variance of effects of diabetes, systolic blood pressure, and their interaction.

	df	MS	F-Statistic	*p*-Value
T2DM	1	64.86	6.5	0.01
SBP	1	92.27	9.25	0.003
T2DM * SBP	1	8.12	0.89	0.368
Standard error	42	9.96		

* On global longitudinal strain. df: degree of freedom; MS: mean square; SBP: systolic blood pressure; T2DM: type 2 diabetes.

**Table 4 ijerph-18-06553-t004:** Two-way analysis of variance of effects of diabetes, overweight, and their interaction * on global longitudinal strain.

	df	MS	F-Statistic	*p*-Value
T2DM	1	67.04	6.50	0.01
Overweight	1	39.90	3.77	0.05
T2DM * Overweight	1	5.77	0.56	0.45
Standard error	196	10.30		

df: degree of freedom; MS: mean square; T2DM: type 2 diabetes.

**Table 5 ijerph-18-06553-t005:** Independent factors of lower GLS risk resulting from logistic regression in hypertensive patients. Adjusted odds ratio (OR) and associated 95% confidence interval (CI).

Risk Factor	OR	95% CI	p^b^
Non control of BP	8.8	2.3–34.2	0.001
BMI	0.2	0.04–1.04	0.005
T2DM	2.26	1.11–4.61	0.023
RWT	5.18	1.84–6.02	0.001

p^b^: variable significance (Wald test).

**Table 6 ijerph-18-06553-t006:** Impact of treatment of HBP on GLS outcome.

	Non Diabetics			Diabetics		
GLS	Patients, *n*	Normal, %	Reduced, %	Patients, *n*	Normal, %	Reduced, %
ACEI	Without, 72	43	56	Without, 78	20.5	79.5
	With, 35	71	28.5	With, 15	53.3	46.3
D	Without, 60	61	38.3	Without, 39	30.7	46.7
	With, 47	40.4	59.6	With, 54	22.2	72.7
BB	Without, 84	55.9	44	Without, 77	28.5	71.4
	With, 23	39.1	61.8	With, 16	12.5	87.5
CCA	Without, 79	55.6	44.3	Without, 59	28.8	71.1
	With, 28	42.8	57.1	With, 34	20.5	79.4

ACEI: angiotensin-converting enzyme inhibitors; BB: beta-blocking agents; CCA: calcium channel antagonists; D: diuretics; GLS: global longitudinal strain.

## Data Availability

Data are available on the dept of cardiology (Pr S Benkhedda), Hospital Mustapha; Algiers.

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
