# Peer review of "Impact of Diabetes on Cardiac Function in Patients with High Blood Pressure"

_ijerph, 2021, doi:10.3390/ijerph18126553_

Round 1

Reviewer 1 Report

In this manuscript,  SOUFI TALEB BENDIAB et al., aimed to find out the impact of T2DM on LV geometry and subclinical dysfunction in hypertensive patients with normal ejection fraction. By using echocardiography coupled with LV GLS assessment, the authors concluded that LV geometry and subclinical LV function are more impaired in hypertensive patients with T2DM than those without T2DM.

    Overall, this single centered, cross-sectional study was well designed. The methods used were appropriate and the results were clearly presented. Although, a similar study was published recently (ref 21), that one used another method (MRI) in the evaluation of cardiac functions. Thus, the same conclusion between these two studies in turn emphasized the validation of the findings. In addition, this reviewer has the following questions:

Major:

  • Compare with the interesting findings, the preventive implications discussed in the section 5.1 were weak and lack of novelty since a lot of study has advocated the importance of controlling body weight or BMI in health management of hypertensive patients or diabetic patients. It will be very interesting if the author could discuss the effectiveness or correlation of the medications used by the patients with T2DM during this study with the LV functional outcome.
  • Any information of the follow up study from the same cohort on HFpEF incidence?

Minor:

  • Some of words in the manuscript has bigger size than the others, e.g. the author names, “indexed left atrial volumes” in the abstract.
  • In table 1, the M/F value was strange, do the authors want to show the M/F ratio? It seems not the exact male or female number. Why not show the percentage directly?
  • Please add ethnicity information of the patients.

Author Response

lls suggestions of the reviewer have been taken into account and appear highlighted in red in the revised submission. Especially, we added a Table 6  with the impact of HBP treatment on GLS outcome.

Reviewer 2 Report

This MS explores the impact on subclinical cardiac function of concomitant diabetes in a cohort of patients with HBP. It is well written and easy to follow. Minor comments:

L.27 - were more impaired

L.76 – Blood samples were used for glucose and creatinine levels ? Please rephrase the sentence: Blood sampling included blood glucose and creatinin levels.

Please consider the following papers:

Rajbhandari J, Fernandez CJ, Agarwal M, Yeap BXY, Pappachan JM. Diabetic heart disease: A clinical update. World J Diabetes. 2021 Apr 15;12(4):383-406. doi: 10.4239/wjd.v12.i4.383.

Petrie JR, Guzik TJ, Touyz RM. Diabetes, Hypertension, and Cardiovascular Disease: Clinical Insights and Vascular Mechanisms. Can J Cardiol. 2018 May;34(5):575-584. doi: 10.1016/j.cjca.2017.12.005.

Author Response

All suggestions of the reviewer have been taken into account. The 2 references have been added too.